# RadiomiX for Radiomics Analysis: Automated Approaches to Overcome Challenges in Replicability

**DOI:** 10.3390/diagnostics15151968

**Published:** 2025-08-05

**Authors:** Harel Kotler, Luca Bergamin, Fabio Aiolli, Elena Scagliori, Angela Grassi, Giulia Pasello, Alessandra Ferro, Francesca Caumo, Gisella Gennaro

**Affiliations:** 1Breast Radiology, Veneto Institute of Oncology IOV-IRCCS, 35128 Padua, Italy; 2Department of Mathematics, University of Padova, 35128 Padua, Italy; 3Radiology, Veneto Institute of Oncology IOV-IRCCS, 35128 Padua, Italy; 4Clinical Research Unit, Veneto Institute of Oncology IOV-IRCCS, 35128 Padua, Italy; 5Medical Oncology 2, Veneto Institute of Oncology IOV-IRCCS, 35128 Padua, Italy; 6Department of Surgery Oncology and Gastroenterology, University of Padova, 35128 Padua, Italy

**Keywords:** radiomics, lung cancer, breast cancer, hepatic encephalopathy, machine learning, replicability

## Abstract

**Background/Objectives:** To simplify the decision-making process in radiomics by employing RadiomiX, an algorithm designed to automatically identify the best model combination and validate them across multiple environments was developed, thus enhancing the reliability of results. **Methods**: RadiomiX systematically tests classifier and feature selection method combinations known to be suitable for radiomic datasets to determine the best-performing configuration across multiple train–test splits and K-fold cross-validation. The framework was validated on four public retrospective radiomics datasets including lung nodules, metastatic breast cancer, and hepatic encephalopathy using CT, PET/CT, and MRI modalities. Model performance was assessed using the area under the receiver-operating-characteristic curve (AUC) and accuracy metrics. **Results:** RadiomiX achieved superior performance across four datasets: LLN (AUC = 0.850 and accuracy = 0.785), SLN (AUC = 0.845 and accuracy = 0.754), MBC (AUC = 0.889 and accuracy = 0.833), and CHE (AUC = 0.837 and accuracy = 0.730), significantly outperforming original published models (*p* < 0.001 for LLN/SLN and *p* = 0.023 for MBC accuracy). When original published models were re-evaluated using ten-fold cross-validation, their performance decreased substantially: LLN (AUC = 0.783 and accuracy = 0.731), SLN (AUC = 0.748 and accuracy = 0.714), MBC (AUC = 0.764 and accuracy = 0.711), and CHE (AUC = 0.755 and accuracy = 0.677), further highlighting RadiomiX’s methodological advantages. **Conclusions:** Systematically testing model combinations using RadiomiX has led to significant improvements in performance. This emphasizes the potential of automated ML as a step towards better-performing and more reliable radiomic models.

## 1. Introduction

Radiomics is an intriguing field in medical imaging, involving the extraction of extensive quantitative data (features) from medical images using advanced computational techniques [1,2]. Its integration with artificial intelligence (AI) has shown potential in advancing medical care toward personalized medicine [3,4,5,6,7].

To effectively explore these opportunities, it is essential to analyze the radiomics data by creating a radiomics pipeline. This involves dividing the data into “train” and “test” subsets (train-test split) and building a machine learning (ML) model, which includes both identifying the most relevant features (feature selection) and using a classifier to categorize the data based on the identified features. The model’s performance can be measured using various performance metrics, such as accuracy and the area under the receiver operating characteristic curve (AUC). These measurements help determine the “best model”, which maximizes the performance metric while minimizing its variability [8,9].

Finding the best model and assuring the reliability of its performance are two of the main challenges in developing a robust radiomics pipeline, which stem directly from the inherent characteristics of the dataset. Radiomics datasets typically have many features paired with a small sample size [10]. This contradicts the recommended ratio of having 5–10 times as many samples as features for effective ML analysis in radiomics [11]. This disparity impedes model performance and increases the risk of unreliable results and overfitting [12]. To enhance the reliability and replicability of the analysis results, the pipeline can be constructed with multiple train–test splits, selecting the best model based on the average performance across all splits [13,14]. In addition, techniques such as cross-validation (CV) are used to mitigate overfitting [10].

In the search for the best model, identifying the best combination of feature selection methods and classifiers is a significant challenge. This is due to the abundance of feature selection methods and classifiers coupled with the absence of standardized models, a complexity that often drives ad hoc selections with non-reproducible results [9]. A relevant approach for tackling this challenge is by using automated ML, an automation algorithm used to streamline the evaluation of each combination aimed at finding the best fitting one.

To address these challenges, we developed RadiomiX, an open-source Python-based automated ML algorithm for systematic testing and evaluation of radiomic models.

RadiomiX systematically explores and assesses a predefined set of models across multiple train–test splits and K-fold CV to identify the most suitable one for a radiomics dataset and ensure higher replicability.

In this paper we validated RadiomiX’s capabilities across four publicly available datasets, each characterized by a large number of features and a limited number of samples.

## 2. Materials and Methods

Due to the retrospective design of the study and the use of anonymized public datasets, this study was deemed exempt from review by the Institutional Ethics Committee.

### 2.1. Algorithm Description

RadiomiX is an automated, open-source, Python-based algorithm designed to identify and optimize the best performing model configuration for a radiomics dataset. It achieves its purpose by systematically evaluating all combinations of a predefined set of feature selection methods and ML classifiers across multiple train–test splits and CV tests, as illustrated in Figure 1.

The predefined combinations include preprocessing scaling methods, feature selection methods, oversampling methods, and ML classifiers, as illustrated in Figure 2.

The preprocessing step includes MinMax, standard and robust scalers used to scale the dataset, as well as an option without scaling.

The predefined set of feature selection methods includes four individual methods (correlation; minimum redundancy and maximum relevance—mRMR; least absolute shrinkage and selection operator—LASSO; and random forest—RF) as well as all combinations of correlation, mRMR, and LASSO. For mRMR and the RF, the number of features must be fixed in advance and was set to 1, 2, 5, 10, 15, and 25.

Two oversampling methods, RandomOverSampler and Synthetic Minority Oversampling Technique (SMOTE), are included to increase the representation of the minority class in the training subset by 0%, 25%, 50%, 75%, 90%, and 100% of the difference from the majority class [15].

The predefined set of seven ML classifiers includes logistic regression (LR), a support vector machine (SVM), RF, k-nearest neighbors (KNNs), Naive Bayes (NB), adaptive boosting (AdaBoost), and extreme gradient boosting (XGBoost), known for their common use and high performance in radiomics tasks [9,16,17,18].

All combinations of feature selection methods and classifiers are evaluated over 10 train–test splits and 5-fold CV for datasets smaller than 100 samples or 10-fold CV for larger datasets [19,20].

RadiomiX systematically trains and tests the model over 10 train–test splits of the datasets, following Bouckaert and Frank’s principles for improving replicability in learning algorithms [21]. This strategy mitigates the risk of selection bias inherent in single train–test splits, as demonstrated in Figure 3. The figure illustrates the feature selection process for categorizing classifying tumors into malignant and benign. It reveals how single-split reliance might overestimate the importance of certain features due to dataset subset composition, such as size or roundness. It also highlights that the multiple-split strategy allows for the recognition of features that prove significant across all splits, such as texture in this example.

The optimization of hyperparameters is outside the scope of this study. Comprehensive details regarding feature selection methods, classifiers, and their respective hyperparameters can be found in Appendix A.

### 2.2. Datasets

RadiomiX was applied to four public retrospective radiomics datasets: two large lung cancer CT datasets and two smaller datasets, one for breast cancer using [^18^F]-FDG PET/CT and another for hepatic encephalopathy using T1-weighted MRI.

The first dataset, reported as the large lung nodule (LLN) dataset, includes 838 CT scans collected from 333 patients diagnosed with large lung nodules (≥15 mm) between July 2020 and April 2022 [22]. The second dataset, reported as the small lung nodule (SLN) dataset contains 736 CT scans from 633 patients with small (5–15 mm) solid lung nodules acquired between January 2007 and December 2018 [23].

The nodules in the LLN and SLN datasets were assigned a binary classification label, categorizing them as either malignant or benign.

The third dataset, reported as the metastatic breast cancer (MBC) dataset, comprises 96 paired baseline-and-follow-up [^18^F]-FDG PET/CT scans collected from 48 patients with metastatic breast cancer treated at Vienna General Hospital between 2010 and 2015 [24]. Across these studies, 228 metastatic lesions were delineated, and each lesion was given a binary label, identifying it as either a metabolic responder (complete or partial response) or a non-responder (stable or progressive disease).

The fourth dataset, reported as the cirrhosis and hepatic encephalopathy (CHE) dataset, comprises 124 brain MRI examinations from 70 cirrhotic patients and 54 non-cirrhotic controls obtained on a scanner between October 2018 and February 2020 [25]. Within the cirrhotic subgroup, 38 show chronic hepatic encephalopathy (HE), with 22 graded ≥2 and the remainder graded 1. Consequently, every scan is annotated for a binary task (HE present vs. absent) and a three-level grading task (none, grade 1, and grade ≥2).

### 2.3. Statistical Analysis

For each dataset, a RadiomiX pipeline was built for testing the predefined 4704 model combinations (4 scaling methods × 12 oversampling methods × 14 feature selection methods × 7 classifiers) and evaluating them over 10 train–test splits and 10-fold CV. The evaluation focused on two performance metrics: AUC and accuracy.

Each model’s average performance metric and 95% confidence interval (CI) were derived by RadiomiX from the models’ performance distribution across all splits and folds. Models achieving average performances ≥ 0.7 in both metrics were considered well-performing.

Out of the 4704 combinations tested, 31 combinations were found to be well-performing for the LLN dataset, 35 for the SLN dataset, 48 for the MBC dataset, and 29 for the CHE dataset.

These well-performing models were then used to generate heatmaps of their components to highlight the best-performing one. The best model was selected based on its highest average score in AUC and accuracy.

The original, previously published pipelines for each dataset were replicated using RadiomiX according to their reported description, including the number of data splits, feature selection methods, classifiers, and performance metrics. To assess the variability in performance, each model was subsequently re-evaluated across 10 distinct train–test splits.

For the MBC dataset, the replicated pipeline corresponded to the one originally reported to achieve the highest accuracy, as we were unable to reproduce the performance of the model claimed to yield the best AUC despite extensive efforts. Moreover, since the original publications for both the MBC and CHE datasets reported accuracy without accompanying confidence intervals, our results are based on replication using the originally described data distribution.

Finally, for each dataset, the performance of the best model and the re-evaluated model across 10 splits was compared to that of the original pipeline using the Mann–Whitney U test. *p*-values less than 0.05 were considered statistically significant.

Python 3.9.16 was used for all analyses. Input and use description, as well as the optimized configurations of the models selected for each task, are presented in Appendix A. The execution time of the whole pipeline, including the optimization, took around 6 h for the LLN and SLN datasets and around 3 h for the MBC and CHE dataset on a machine with 16 GB of RAM and 10-core Intel Xeon 4310 processors. Statistical analyses were performed by OriginPro 2024 (OriginLab Corporation, Northampton, MA, USA) and MedCalc Statistical version 22.021 (MedCalc Software Ltd., Ostend, Belgium).

## 3. Results

### 3.1. Dataset Characteristics

Table 1 shows the characteristics of the public datasets analyzed by RadiomiX, including the classification label used in this work, the number of patients/samples (overall and for each label category), the number of initial features, the number of train–test splits, and the ratio of each these splits. The LLN and SLN datasets were bigger in both sample size and feature number than the MBC and CHE datasets. Furthermore, the first two datasets use a train–test split ratio of 70:30, while the MBC dataset uses a ratio of 80:20 [22,23,24]. The CHE dataset had a class imbalance with twice as many samples without HE (HE absent = 86) than samples with HE (HE present = 38), and its train–test split ratio was not published in the original work [25].

### 3.2. RadiomiX Performance

Table 2 summarizes the comparative performance analysis between the best models selected by RadiomiX and the originally published results across four distinct radiomics datasets.

RadiomiX demonstrated significantly better performance in lung nodule malignancy classification in both the LLN and SLN datasets. In the LLN dataset, RadiomiX led to an improvement in the AUC from 0.83 to 0.850 (*p* < 0.001) and accuracy from 0.76 to 0.785 (*p* < 0.001) over the original methodology. Similarly, in the SLN dataset the AUC improved from 0.78 to 0.845 (*p* < 0.001) and accuracy from 0.73 to 0.754 (*p* < 0.001) compared to the original model.

Response classification based on the MBC dataset evaluation revealed mixed but encouraging results. RadiomiX achieved a statistically significant accuracy improvement from 0.76 to 0.833 (*p* = 0.023) while maintaining non-inferior discriminative performance with the AUC increasing from 0.85 to 0.889 (*p* = 0.190).

The results of the HE presence classification in the CHE dataset further validated RadiomiX’s automated approach, demonstrating non-inferiority to the original manual model. RadiomiX achieved comparable discriminative performance with an AUC of 0.837 versus the original 0.82 (*p* = 0.530) and maintained equivalent accuracy at 0.730 compared to the original 0.729 (*p* = 0.928).

RadiomiX’s 10-split 10-fold CV method led to greater variability in performance than the original published results in all four datasets, which used a single train–test split in their pipeline, as illustrated in Figure 4.

### 3.3. The Model’s Performance Is Dependent on the Dataset

Figure 5 shows heatmaps illustrating the average performance of the well-performing model combinations as calculated by RadiomiX for each dataset. These combinations are composed of five distinct feature selection methods or their combinations paired with seven classifiers. The color intensity in the heatmaps corresponds to the performance level of the model, where higher intensity indicates higher performance. Each heatmap was intentionally scaled independently to highlight the subtle but important differences in performance between the combinations of feature selection methods and classifiers within each dataset. Adopting a unified color scale across all heatmaps would have compressed these variations, potentially obscuring important contrasts, particularly in datasets where performance values are closely clustered. Furthermore, the white cells represent model combinations (i.e., feature selection method and classifier pairs) that did not reach a predefined performance threshold and were thus excluded from visualization to reduce visual noise and emphasize the most relevant results.

For the LLN and MBC datasets, RadiomiX identified single-model combinations that achieved the best performance for both the AUC and accuracy metrics.

In the LLN dataset’s lung nodule classification task, RadiomiX identified the best combination as RF for both feature selection and classification (AUC = 0.850 and 95% CI = 0.734, 0.919; accuracy = 0.785 and 95% CI = 0.694, 0.863). In the MBC dataset, the best models identified by RadiomiX for the response classification task was based on LASSO feature selection coupled to an RF classifier (AUC = 0.889 and 95% CI = 0.768, 0.979; accuracy = 0.833 and 95% CI = 0.714, 0.952).

In contrast, the SLN and CHE datasets revealed more complex landscapes where RadiomiX identified distinct model combinations for the optimal AUC versus accuracy performance. For the lung nodule classification task in the SLN dataset, the highest performance in the AUC emerged from the multi-method feature selection approach, combining mRMR, correlation, and LASSO methods with an LR classifier (AUC = 0.845 and 95% CI = 0.772, 0.915). Conversely, the highest accuracy was achieved through LASSO feature selection paired with the SVM classifier (accuracy = 0.771 and 95% CI = 0.676, 0.851). Similarly, the HE presence classification task of the CHE dataset exhibited metric-specific best combinations. The highest AUC was achieved through LASSO feature selection with an RF classifier (AUC = 0.837 and 95% CI = 0.649, 0.967), while best accuracy performance required the same multi-method feature selection incorporating mRMR, correlation, and LASSO approaches combined with an RF classifier (accuracy = 0.758 and 95% CI = 0.545, 0.900).

### 3.4. Performance Dependence on Dataset Train–Test Splitting

The evaluation of original model architectures under multiple splits and CV revealed important insights into the robustness and generalizability. Table 3 presents the comparison between the originally reported performance metrics and their recalculated equivalents when subjected to 10 independent train–test splits, providing a more comprehensive assessment of model stability.

Across all four datasets, the 10-split replication approach exposed performance variability that was not apparent in single-split evaluations. In the lung nodule classification datasets, the 10-split LLN model replication showed significantly lower performance, with the AUC declining from 0.83 to 0.783 (*p* < 0.001) and accuracy from 0.76 to 0.731 (*p* < 0.001). Similarly, the 10-split replication of the original SLN model showed an AUC reduction from 0.78 to 0.748 (*p* < 0.001) and an accuracy decrease from 0.73 to 0.714 (*p* < 0.001).

The 10-split recalculation of the response classification task based on the MBC dataset revealed a significant decrease in AUC, declining from 0.85 to 0.764 (*p* = 0.005). While the accuracy showed a decrease from 0.763 to 0.711, it did not reach statistical significance (*p* = 0.064). Similarly, the HE-presence classification task in the CHE dataset demonstrated substantial performance dependencies on train–test composition, where replication of 10-split model achieved a significant decrease in AUC from 0.82 to 0.755 (*p* = 0.003) and a decline in accuracy from 0.729 to 0.677 (*p* = 0.252).

When contextualized against RadiomiX’s performance presented in Table 2, the performance gaps between the original single–split results and their 10-split recalculations highlight that RadiomiX’s improvements become even more pronounced when compared against methodologically equivalent baselines across all tasks (*p* < 0.001).

The box plots presented in Figure 6 demonstrate that performance variability is greater under the 10-train–test-split framework compared to the originally used single-split evaluations across all four datasets.

## 4. Discussion

In this study, we introduced RadiomiX, an automatic machine learning modular algorithm that aims to identify the best model for radiomic datasets by extensive testing and evaluation on multiple train–test splits. We validated RadiomiX on four public retrospective datasets. For each dataset, the algorithm automatically found the best-performing model that outperformed those previously reported. Each of the four best-performing models was composed of a different combination of feature selection methods and classifiers. In addition, two of the best-performing models were based on a combination of features selection methods (mRMR, correlation, and LASSO), demonstrating the potential of combined methods despite being used in only a few works in recent years [26,27,28,29].

These results not only reinforce the common assumption that no model is universally applicable for all tasks [30,31] but also show the potential of automated ML in overcoming the hurdle of finding the most suitable one. The results obtained with RadiomiX are consistent with recent advances in automated ML such as the TPOT [32] and AutoPrognosis [33] frameworks, which demonstrated how automated methods can overcome manual efforts in identifying effective models in large medical datasets.

In the SLN and CHE datasets, the best model was different depending on the optimization metric, AUC or accuracy. This emphasizes the importance of choosing the correct metric for optimization and performance evaluation, in line with the clinical objectives of the task, as different metrics may also lead to different model selection. The clinical rationale behind the choice of optimization metrics should be carefully considered during radiomics study design and presented along with the development and optimization process.

Replication of the originally published single-split models in multiple train–test splits demonstrated significantly reduced performance. This result reinforces recent work by An et al. on the unreliability of a single train–test split in small datasets. Their research shows that model performance can vary significantly between different pairs of training–test sets, and they suggest the adoption of bagging and CV techniques to mitigate this phenomenon [13]. This concept led to the incorporation of multiple train–test splits into the RadiomiX algorithm, making it original in the field of automated ML.

In designing RadiomiX, we concentrated on the three pipeline components: the classifier choice, the FS method, and the train–test split strategy. These components have been shown by Decoux et al. to exert the greatest influence on radiomic model performance after the intrinsic quality of the dataset itself, quantified at >20% of the total performance variance observed across published radiomics studies [17].

We also adopted a 90:10 train–test split, whereas the original benchmark papers used ratios between 70:30 and 80:20. Rácz et al. showed that, in the small-sample settings typical of radiomics, reducing the size of the held-out set has only a marginal impact on the AUC and accuracy [34]. Consequently, the performance improvements attributed to RadiomiX stem primarily from its targeted selection of classifier and FS combinations rather than from any favorable alteration of the evaluation protocol, more so considering the multiple train–test environment.

Another methodological consideration in RadiomiX design involves the heterogeneity in feature extraction approaches across datasets. While the LLN and SLN datasets used 25 HU binning with TexLab 2.0, MBC and CHE employed Image Biomarker Standardization Initiative (IBSI)-compliant features via LifeX and PyRadiomics, respectively [2,22,23,24,25]. We selected this approach to make RadiomiX’s extraction agnostic, increasing practical utility in real-world scenarios where researchers work with diverse radiomic datasets using different extraction pipelines and varying IBSI compliance levels.

This study is not without limitations. First, the publicly available data did not include external datasets in all cases, so the best-performing models were not validated on external datasets or prospective data.

Second, only four datasets were used, and for each only a single classification label was used. This hinders the understanding of whether the best-performing model is dependent on the dataset or the task. Future works should analyze more datasets with various classification labels and in a broader range of clinical scenarios.

Third, the analyzed datasets were already prepared and ready for processing. This suggests that preliminary work should be performed on an extracted unorganized dataset before applying the algorithm to it.

To further improve RadiomiX, future works could focus on adding more feature selection methods and cover steps out of the scope of this study such as feature extraction.

A more comprehensive hyperparameter optimization method could also improve performance. While we used limited grid search with reduced parameter values below the automatic defaults to minimize overfitting in small radiomic datasets, extensive optimization could yield further improvements.

Feature selection methods, such as symmetric uncertainty (SUMeasure) and information gain filtering methods, as well as Boruta, a high-stability high-performance wrapper method, have demonstrated their effectiveness in radiomics and could be implemented in RadiomiX [8].

Feature extraction could be implemented by incorporating pyRadiomics.

Hyperparameter optimization could be included by using techniques that have already shown promise in automated ML such as Auto-Sklearn [35], Bayesian optimization, or genetic algorithms.

## 5. Conclusions

This study introduces RadiomiX, an automated ML algorithm suitable for radiomics datasets and aimed at finding the best model for them. Since it evaluates each model over multiple train–test splits, RadiomiX aspires to ensure greater replicability of the selected model. When validated on previously published datasets, RadiomiX automatically selected a distinct model for each of the datasets that outperformed the previously published models reported as the best models. This evidence reinforces the potential of automated ML in finding task- and dataset-specific models in hopes of obtaining better-performing and more reliable radiomic models.

## Figures and Tables

**Figure 1 diagnostics-15-01968-f001:**
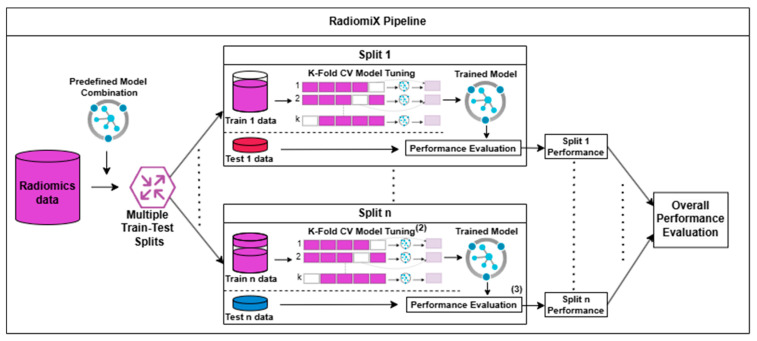
RadiomiX pipeline. The RadiomiX pipeline performs the following steps on radiomic data when training and testing a predefined model combination: first, the data are split into multiple train and test data subsets, where each subset is treated separately. Second, each train data subset is used for K-Fold cross-validation to train and tune the predefined model combination. Third, each split-trained model’s performance is evaluated on the same split test data subset. Finally, the overall performance of the model is calculated based on its performance distribution across all splits. CV, cross-validation.

**Figure 2 diagnostics-15-01968-f002:**
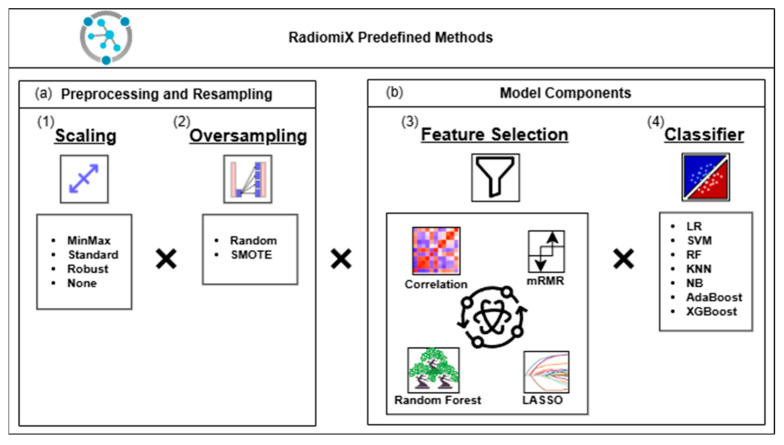
RadiomiX’s predefined methods included in the analysis. RadiomiX evaluated all the model combinations of the following methods: (**a**) preprocessing and resampling methods: (1) Scaling methods: MinMaxScaler, StandardScaler, RobustScaler, and no scaling; (2) oversampling methods applied to the training subset after feature selection: RandomOverSampler and SMOTE; (**b**) model components: (3) feature selection methods: correlation, mRMR, LASSO, RF, and all the combinations of correlation, mRMR, and LASSO; (4) classifiers: LR, SVM, RF, KNN, NB, AdaBoost, and XGBoost. AdaBoost, Adaptive Boosting; KNN, K-Nearest Neighbors; LASSO, Least Absolute Shrinkage and Selection Operator; LR, Logistic Regression; mRMR, Minimum Redundancy and Maximum Relevance; MinMax, Minimum Maximum; NB, Naive Bayes; RF, Random Forest; SMOTE, Synthetic Minority Oversampling Technique; SVM, Support Vector Machine; XGBoost, Extreme Gradient Boosting.

**Figure 3 diagnostics-15-01968-f003:**
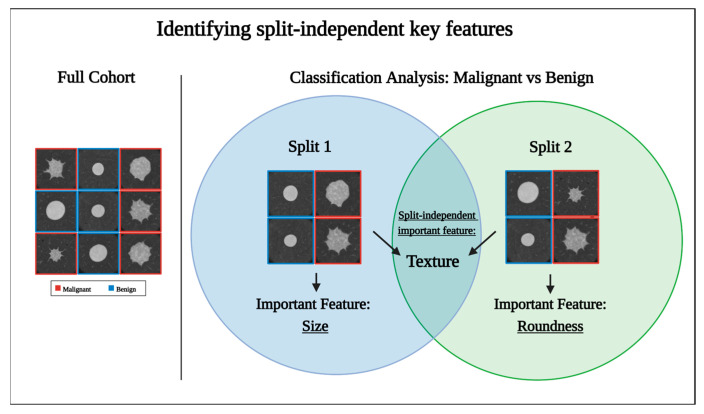
The selection process of split-independent key features for classifying mock tumors into malignant and benign. The full cohort of tumors is presented on the left side, and two train–test splits of that cohort are presented on the right. As the selected features are greatly affected by the samples used for analysis, each split led to the identification of different important features. This illustrates the limitation of relying on a singular data split for feature selection, where certain attributes like size or shape might misleadingly seem important due to the specific composition of the dataset subset. To address this issue, using multiple data splits allows for the effective identification of features that consistently demonstrate importance across all splits, such as texture.

**Figure 4 diagnostics-15-01968-f004:**
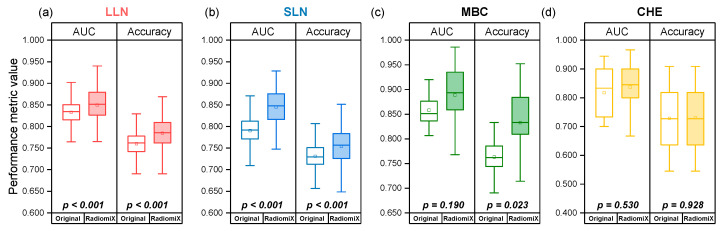
Box plot distributions comparing the originally published performance against RadiomiX’s performance. (**a**) LLN AUC and accuracy; (**b**) SLN AUC and accuracy; (**c**) MBC AUC and accuracy; (**d**) CHE AUC and accuracy. The box represents the interquartile range (25th to 75th percentile), the horizontal line indicates the median, and the square denotes the mean. AUC, area under the receiver operating characteristic curve; CHE, cirrhosis and hepatic encephalopathy; LLN, large lung nodule; MBC, metastatic breast cancer; SLN, small lung nodule.

**Figure 5 diagnostics-15-01968-f005:**
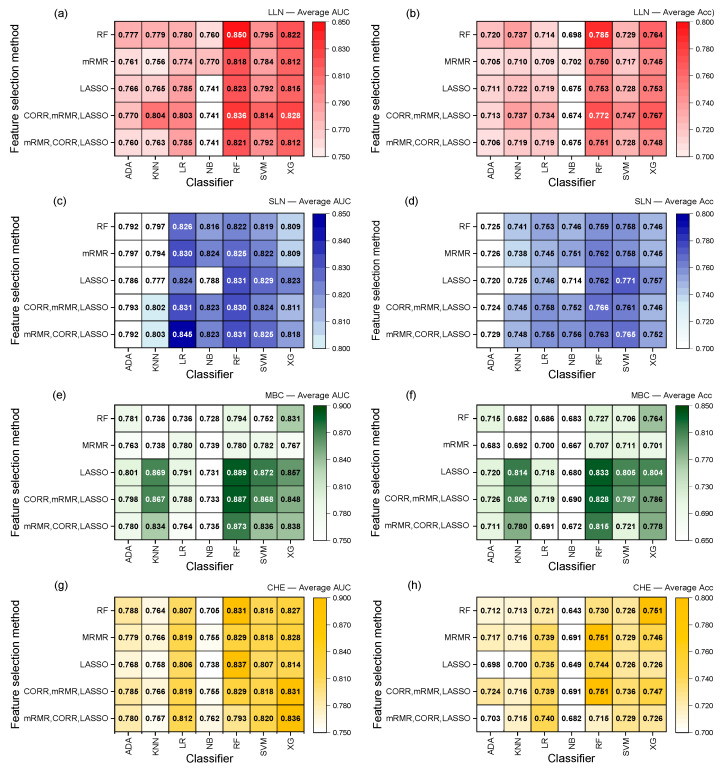
Feature selection method and classifier combination heatmaps for each public dataset and metric: (**a**) LLN dataset—average AUC; (**b**) LLN dataset—average accuracy; (**c**) SLN dataset—average AUC; (**d**) SLN dataset—average accuracy; (**e**) MBC dataset—average AUC; (**f**) MBC dataset—average accuracy; (**g**) CHE dataset—average AUC; (**h**) CHE dataset—average accuracy. Combined methods are separated by commas in the order of their application. Each heatmap was intentionally scaled independently, and white cells represent feature selection method and classifier pairs that did not reach a predefined performance threshold and were thus excluded from visualization. Acc, accuracy; AUC, area under the receiver operating characteristic curve; CHE, cirrhosis and hepatic encephalopathy; CORR, correlation; mRMR, maximum relevance—minimum redundancy; LASSO, least absolute shrinkage and selection operator; LLN, large lung nodule; MBC, metastatic breast cancer; SLN, small lung nodule.

**Figure 6 diagnostics-15-01968-f006:**
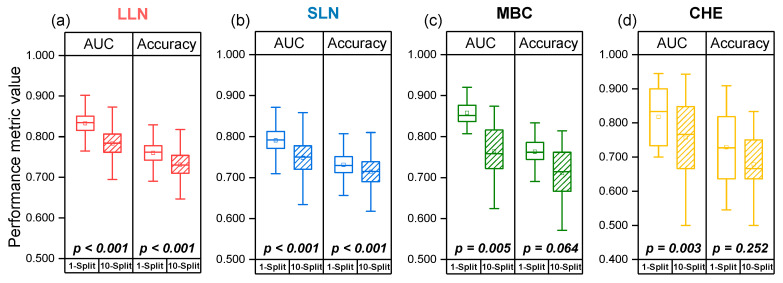
Box plot distributions comparing the original published performance against their recalculated performance across 10 splits. (**a**) LLN AUC and accuracy; (**b**) SLN AUC and accuracy; (**c**) MBC AUC and accuracy; (**d**) CHE AUC and accuracy. The box represents the interquartile range (25th to 75th percentile), the horizontal line indicates the median, and the square denotes the mean. AUC, area under the receiver operating characteristic curve; LLN, large lung nodule; PDVA, Parkinson’s disease voice assessment; SLN, small lung nodule.

**Table 1 diagnostics-15-01968-t001:** Characteristics of the public datasets included in the analysis.

Dataset	Classification Label	N° Samples	N° Features	N° Train-Test Splits (Ratio)
LLN	Malignant/ benign	Total: 838 Malignant: 524 Benign: 314	666 *	1 (70:30)
SLN	Malignant/ benign	Total: 736 Malignant: 377 Benign: 359	1998	1 (70:30)
MBC	Responders/non-responders	Total: 228 Responders: 127 Non-responders: 101	222	1 (80:20)
CHE	HE present/ HE absent	Total: 124 HE present: 38 HE absent: 86	43	1 (Ratio undisclosed)

* In the original paper 666 features were extracted, yet only 82 were available in the publicly available dataset. CHE, cirrhosis and hepatic encephalopathy; HE, hepatic encephalopathy; LLN, large lung nodule; MBC, metastatic breast cancer; N°, number; SLN, small lung nodule.

**Table 2 diagnostics-15-01968-t002:** RadiomiX’s best model classification performance compared with the originally published performance.

Dataset	Performance Metric	Originally Published Performance (95% CI)	RadiomiX’s Best Model Performance (95% CI)	*p*-Value
LLN	AUC	0.83 (0.77, 0.88) *	0.850 (0.734, 0.919)	<0.001
	Accuracy	0.76 (0.70, 0.81) *	0.785 (0.694, 0.863)	<0.001
SLN	AUC	0.78 (0.70, 0.86) *	0.845 (0.772, 0.915)	<0.001
	Accuracy	0.73 (0.65, 0.81) *	0.754 (0.653, 0.830)	<0.001
MBC	AUC	0.85 (0.73, 0.95) *	0.889 (0.768, 0.979)	0.190
	Accuracy	0.763 (0.696, 0.829)	0.833 (0.714, 0.952)	0.023
CHE	AUC	0.82 (0.73-0.90) *	0.837 (0.649, 0.967)	0.530
	Accuracy	0.729 (0.566, 0.892)	0.730 (0.584-0.909)	0.928

The 95% confidence intervals are displayed in parentheses. The number of correctly classified patients/samples is displayed in brackets. The metrics for optimization are underscored. * The performance metric was originally published with only two significant digits. AUC, area under the receiver operating characteristic curve; CI, confidence interval; CHE, cirrhosis and hepatic encephalopathy; LLN, large lung nodule; MBC, metastatic breast cancer; SLN, small lung nodule.

**Table 3 diagnostics-15-01968-t003:** Originally published performance compared with its recalculation over 10 train–test splits.

Dataset	Performance Metric	Originally Published Performance Based on a Single Split (95% CI)	Recalculated Performance Based on 10 Splits (95% CI)	*p*-Value
LLN	AUC	0.83 (0.77, 0.88) *	0.783 (0.717, 0.846)	<0.001
	Accuracy	0.76 (0.70, 0.81) *	0.731 (0.667, 0.794)	<0.001
SLN	AUC	0.78 (0.70, 0.86) *	0.748 (0.668, 0.821)	<0.001
	Accuracy	0.73 (0.65, 0.81) *	0.714 (0.644, 0.781)	<0.001
MBC	AUC	0.85 (0.73, 0.95) *	0.764 (0.626, 0.871)	0.005
	Accuracy	0.763 (0.696, 0.829)	0.711 (0.600, 0.805)	0.064
CHE	AUC	0.82 (0.73, 0.90) *	0.755 (0.533, 0.933)	0.003
	Accuracy	0.729 (0.566, 0.892)	0.677 (0.500, 0.833)	0.252

* The performance metric was originally published with only two significant digits. The 95% confidence intervals are displayed in parentheses. The number of correctly classified patients/samples is displayed in brackets. The metrics for optimization are underscored. AUC, area under the receiver operating characteristic curve; CI, confidence interval; CHE, cirrhosis and hepatic encephalopathy; LLN, large lung nodule; MBC, metastatic breast cancer; SLN, small lung nodule.

## Data Availability

RadiomiX is available at https://github.com/HarelKotler/RadiomiX. The four publicly available datasets come from the Mendeley Data repository, Harvard Dataverse, and Zenodo. The links are as follows: https://data.mendeley.com/datasets/rz72hs5dvg/1, https://data.mendeley.com/datasets/rxn95mp24d/1, https://dataverse.harvard.edu/dataset.xhtml?persistentId=doi:10.7910/DVN/LA8PGN, and https://zenodo.org/records/7615841.

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
