# Peer review of "RadiomiX for Radiomics Analysis: Automated Approaches to Overcome Challenges in Replicability"

_diagnostics, 2025, doi:10.3390/diagnostics15151968_

Round 1
Reviewer 1 Report
Comments and Suggestions for Authors
The paper introduced a significant issue in medical imaging, but it needs an improvement in several issues;
- the introduction needs much improvement , related to the previous studies on using radmoics features in diagnosis.
- mention the radomics features that have been used
- "hoc " mention the full name for this abbreviation.
- show the frame work of your designed system.
- the heat maps are not clear, so I recommend to re generate them.
- what are the types of feature selection methods that have been used here, explain a little bit on each method
- for all types of classifiers talk about them and their equations.
Reviewer 1 Report
The paper introduced a significant issue in medical imaging, but it needs an improvement in several issues;
- The introduction needs much improvement, related to the previous studies on using radiomics features in diagnosis.
- Mention the radiomics features that have been used
- "hoc " mention the full name for this abbreviation.
- Show the framework of your designed system.
- The heat maps are not clear, so I recommend to re-generate them.
- What are the types of feature selection methods that have been used here, explain a little bit on each method
- For all types of classifiers talk about them and their equations.
Author Response
General comment: The paper introduced a significant issue in medical imaging, but it needs an improvement in several issues.
Response: Thank you very much for acknowledging the importance of this topic, please find below our point by point answers to your concerns and the corresponding changes in the current version of the manuscript.
Point 1: the introduction needs much improvement, related to the previous studies on using radmoics features in diagnosis.
Response: We thank the reviewer for their suggestion to include studies on the use of radiomics features in diagnosis. While we acknowledge the importance of such literature, the primary focus of our manuscript is methodological, specifically, on the development and validation of RadiomiX, an automated machine learning pipeline for radiomics analysis. Our intention is not to review the diagnostic applications of radiomics, but rather to address challenges related to model reliability, replicability, and performance assessment in high-dimensional, low-sample-size datasets typical of radiomic studies.
Including an in-depth discussion of clinical diagnostic applications of radiomics would broaden the scope of the introduction beyond our methodological focus and may dilute the clarity of the study's aims. For this reason, we respectfully suggest maintaining the current scope, which centers on the optimization and evaluation of radiomics pipelines.
Point 2: mention the radomics features that have been used
Response: Thank you for this request for additional detail regarding the radiomic features utilized in our study. In response to your suggestion, we have added comprehensive information about all radiomic features used in our analysis to the S2 final table in the supplementary material. This addition provides readers with complete transparency regarding the specific features used.
Point 3: "hoc " mention the full name for this abbreviation.
Response: The term "hoc" in this context is not a standalone abbreviation but part of the complete Latin expression "ad hoc," meaning "for this specific purpose", which is commonly used in academic literature without requiring expansion.
Point 4: show the framework of your designed system.
Response: Thank you for requesting clarification on our system framework. We have added a detailed description of our designed system framework to Supplemental Material S2, which now includes comprehensive information. We placed this information in the supplemental material to maintain the manuscript's readability and focus on the core research findings, while ensuring that readers have full access to the technical implementation details. Additionally, we have made available the complete framework documentation, including installation procedures, execution guidelines, parameter definitions, and detailed usage instructions for RadiomiX, on our GitHub repository for readers who require more comprehensive technical information. If you believe that including a summary of the framework architecture in the main manuscript would better serve the readers' understanding, we would be happy to add this detailed explanation to the appropriate section.
Point 5: the heat maps are not clear, so I recommend to re-generate them.
Response: We appreciate the reviewer’s feedback regarding the clarity of the heatmaps in Figure 5. However, since the specific nature of the concern was not detailed, it is difficult to determine whether the issue pertains to resolution, axis or label legibility, color scale consistency, or insufficient contextual explanation.
In the absence of a more specific indication, we have considered that the use of dataset-specific color scales and the presence of white cells may have contributed to the perceived lack of clarity.
To clarify, each heatmap was intentionally scaled independently to highlight the subtle but important differences in performance between the combinations of feature selection methods and classifiers within each dataset. Adopting a unified color scale across all heatmaps would have compressed these variations, potentially obscuring important contrasts, particularly in datasets where performance values are closely clustered.
Furthermore, the white cells represent model combinations (i.e., feature selection method and classifier pairs) that did not reach a predefined performance threshold and were thus excluded from visualization to reduce visual noise and emphasize the most relevant results.
We have revised the figure legend and main text to more clearly explain both the scaling approach and the use of white cells.
Point 6: what are the types of feature selection methods that have been used here, explain a little bit on each method
Response: In response to your feedback, we have added comprehensive descriptions of each feature selection method employed in our study, including their underlying principles, key equations, and relative advantages and disadvantages. These detailed explanations have been included in the supplemental material S1 to maintain the flow of the main manuscript while providing readers with easy access to in-depth technical information.
Point 7: for all types of classifiers talk about them and their equations.
Response: Following your suggestion, and similarly to the feature selection additional explanation, we have provided comprehensive descriptions of all classifier types, including their underlying mathematical formulations, key equations, and a summary of their strengths and limitations. These detailed explanations have been added to the supplemental material S1 to ensure readers have access to thorough technical information without disrupting the narrative flow of the main paper.
Reviewer 2 Report
Comments and Suggestions for Authors
The manuscript presents RadiomiX, a complex software system designed to identify the optimal model for radiomic datasets. The study aimed to validate the RadiomiX system using four publicly available retrospective datasets. The manuscript presents a promising algorithmic system that is likely to support the practice of radiomic analysis; however, the study raises several questions.
Comments/Remarks:
- The method section provides a sufficiently detailed description of RadiomiX's operating scheme, but it would also be helpful to describe the program architecture (even if only as supplementary material). Is the program command-line-based, or does it offer GUI support for users? How should the program be configured: using multiple text files? How many different commands are needed to run a model analysis? How should the input data be made available: as DICOM files and CSV tables?
- Page 4. "All combinations of feature selection methods and classifiers are evaluated over 10 train-test splits and 5-fold CV, for datasets smaller than 200 samples, or 10-fold CV, for larger datasets [19]". The reference was published in 2005, which does not seem recent enough. Is there any more recent published information or facts regarding the necessary train-test splits and CVs?
- What is the maximum number of train-test splits and CV that can be set in the program?
- Additional details about the feature selection methods, the classifiers, and their respective hyperparameters are in the Supplemental Material. It would be worthwhile to standardize the tables for the three methods (Feature selection, Classifiers and Scalers), as each one includes the name of the method, the hyperparameters, and the implementation method.
- Page 6. "The execution time of the whole pipeline, including the optimization, took around 6 hours per dataset on a machine with 16 GB RAM and 10-core Intel Xeon 4310 processors". The four datasets contained fundamentally different numbers of cases. It would be helpful to include specific running times for the four datasets.
- It is not clear how (in which process) the discretization step was included in the radiomics calculation. The discretization could be an absolute or relative method (fixed bin size or fixed bin number), as defined in https://doi.org/10.48550/arXiv.1612.07003. It has been demonstrated that various quantization methods can significantly influence the calculated radiomic/textural features and, more importantly, the correlational relationships between other variables.
- The specific function form of the radiomics and texture parameters must correspond to the types defined/recommended in the IBSI standard. Did the implementation follow the function definitions in the recommendation (https://doi.org/10.48550/arXiv.1612.07003)? Were all the suggested functions of the IBSI implemented in RadiomiX?
- Page 5. "For each dataset, a RadiomiX pipeline was built for testing the predefined 532 model combinations and evaluating them over 10 train-test splits and 10-fold CV." Please briefly describe why there are a total of 532 different model combinations.
- Table 2 shows the results of the comparative analysis between the best models selected by RadiomiX and the originally published data from the four distinct radiomics datasets. For basic comparison purposes, it would be helpful to see the AUC and accuracy results calculated by RadiomiX, which were determined using exactly the same steps/methods (scalers, feature selection, classifier, and related hyperparameters) as in the original publications.
- The last table in the Supplemental material S2 presents the "optimal pipelines' settings and hyperparameters" for the four different datasets. In the case of the MBC and the CHE, the classifier and feature selection methods were the same; however, the hyperparameters differed. Additionally, the authors emphasized that optimizing hyperparameters was outside the scope of this study. In this regard, why did the hyperparameters differ between the MBC and CHE datasets?
Reviewer 2 Report
The manuscript presents RadiomiX, a complex software system designed to identify the optimal model for radiomic datasets. The study aimed to validate the RadiomiX system using four publicly available retrospective datasets. The manuscript presents a promising algorithmic system that is likely to support the practice of radiomic analysis; however, the study raises several questions.
Comments/Remarks:
- The method section provides a sufficiently detailed description of RadiomiX's operating scheme, but it would also be helpful to describe the program architecture (even if only as supplementary material). Is the program command-line-based, or does it offer GUI support for users? How should the program be configured: using multiple text files? How many different commands are needed to run a model analysis? How should the input data be made available: as DICOM files and CSV tables?
- Page 4. "All combinations of feature selection methods and classifiers are evaluated over 10 train-test splits and 5-fold CV, for datasets smaller than 200 samples, or 10-fold CV, for larger datasets [19]". The reference was published in 2005, which does not seem recent enough. Is there any more recent published information or facts regarding the necessary train-test splits and CVs?
- What is the maximum number of train-test splits and CV that can be set in the program?
- Additional details about the feature selection methods, the classifiers, and their respective hyperparameters are in the Supplemental Material. It would be worthwhile to standardize the tables for the three methods (Feature selection, Classifiers and Scalers), as each one includes the name of the method, the hyperparameters, and the implementation method.
- Page 6. "The execution time of the whole pipeline, including the optimization, took around 6 hours per dataset on a machine with 16 GB RAM and 10-core Intel Xeon 4310 processors". The four datasets contained fundamentally different numbers of cases. It would be helpful to include specific running times for the four datasets.
- It is not clear how (in which process) the discretization step was included in the radiomics calculation. The discretization could be an absolute or relative method (fixed bin size or fixed bin number), as defined in https://doi.org/10.48550/arXiv.1612.07003. It has been demonstrated that various quantization methods can significantly influence the calculated radiomic/textural features and, more importantly, the correlational relationships between other variables.
- The specific function form of the radiomics and texture parameters must correspond to the types defined/recommended in the IBSI standard. Did the implementation follow the function definitions in the recommendation (https://doi.org/10.48550/arXiv.1612.07003)? Were all the suggested functions of the IBSI implemented in RadiomiX?
- Page 5. "For each dataset, a RadiomiX pipeline was built for testing the predefined 532 model combinations and evaluating them over 10 train-test splits and 10-fold CV." Please briefly describe why there are a total of 532 different model combinations.
- Table 2 shows the results of the comparative analysis between the best models selected by RadiomiX and the originally published data from the four distinct radiomics datasets. For basic comparison purposes, it would be helpful to see the AUC and accuracy results calculated by RadiomiX, which were determined using exactly the same steps/methods (scalers, feature selection, classifier, and related hyperparameters) as in the original publications
- The last table in the Supplemental material S2 presents the "optimal pipelines' settings and hyperparameters" for the four different datasets. In the case of the MBC and the CHE, the classifier and feature selection methods were the same; however, the hyperparameters differed. Additionally, the authors emphasized that optimizing hyperparameters was outside the scope of this study. In this regard, why did the hyperparameters differ between the MBC and CHE datasets?
Author Response
General comment: The manuscript presents RadiomiX, a complex software system designed to identify the optimal model for radiomic datasets. The study aimed to validate the RadiomiX system using four publicly available retrospective datasets. The manuscript presents a promising algorithmic system that is likely to support the practice of radiomic analysis; however, the study raises several questions.
Response: Thank you very much for recognizing the potential of RadiomiX as a promising algorithmic system for supporting radiomic analysis practice. We greatly appreciate your thorough and insightful review - it is evident from your detailed comments and thoughtful questions that you possess in-depth knowledge of the radiomic field, which makes your feedback particularly valuable to us. We have carefully addressed each of the questions you raised and have made corresponding improvements to strengthen the manuscript. Please find below our detailed point-by-point responses to your concerns, along with references to the specific sections and line numbers in the current version of the manuscript where these issues have been addressed.
Point 1: The method section provides a sufficiently detailed description of RadiomiX's operating scheme, but it would also be helpful to describe the program architecture (even if only as supplementary material). Is the program command-line-based, or does it offer GUI support for users? How should the program be configured: using multiple text files? How many different commands are needed to run a model analysis? How should the input data be made available: as DICOM files and CSV tables?
Response: We have added a detailed description to Supplemental Material S2. We placed this information in the supplemental material to maintain the manuscript's readability and focus on the core research findings, while ensuring that readers have full access to the technical implementation details. Additionally, we have made available the complete framework documentation, including installation procedures, execution guidelines, parameter definitions, and detailed usage instructions for RadiomiX, on our GitHub repository for readers who require more comprehensive technical information. If you believe that including a summary of the framework architecture in the main manuscript would better serve the readers' understanding, we would be happy to add this detailed explanation to the appropriate section.
Point 2: Page 4. "All combinations of feature selection methods and classifiers are evaluated over 10 train-test splits and 5-fold CV, for datasets smaller than 200 samples, or 10-fold CV, for larger datasets [19]". The reference was published in 2005, which does not seem recent enough. Is there any more recent published information or facts regarding the necessary train-test splits and CVs?
Response: We appreciate your attention to ensuring that our methodological choices are supported by the most recent evidence. In response to your concern, we have updated our approach based on more recent literature. Specifically, we have incorporated guidance from Tsamardinos et al. (2022), which provides updated recommendations for cross-validation strategies based on sample size. Following this more recent reference, we have revised our threshold from 200 samples to 100 samples, with datasets smaller than 100 samples using 5-fold CV and larger datasets using 10-fold CV. We note that this revision did not affect our actual analysis outcomes, as the MBC dataset analysis was performed with 10-fold CV (having 228 samples, which exceeded both the original 200 and revised 100 sample thresholds).
The following updates were made:
- Manuscript line 124 (Materials and Methods -> Algorithm Description) - Changed the threshold from 200 to 100.
- Manuscript line 125 (Materials and Methods -> Algorithm Description) - added the relevant citation (in the next point)
- Manuscript line 461 (citations) - 20. Tsamardinos I (2022) Don’t lose samples to estimation. Patterns 3:100612. https://doi.org/10.1016/j.patter.2022.100612
Point 3: What is the maximum number of train-test splits and CV that can be set in the program?
Response: RadiomiX does not impose a maximum limit on the number of train-test splits or cross-validation folds that can be configured in the program. Users have the flexibility to define k=n if desired, allowing for maximum utilization of available data. However, given that radiomic datasets are typically characterized by small sample sizes, we did not specify a maximum threshold for k in our methodology description, as the practical constraints are naturally imposed by the dataset size itself. This design choice provides researchers with the flexibility to adapt the validation strategy to their specific dataset characteristics and research requirements.
Point 4: Additional details about the feature selection methods, the classifiers, and their respective hyperparameters are in the Supplemental Material. It would be worthwhile to standardize the tables for the three methods (Feature selection, Classifiers and Scalers), as each one includes the name of the method, the hyperparameters, and the implementation method.
Response: You are absolutely correct that having a consistent format across all methodological tables would significantly improve clarity and usability for readers. In response to your feedback, we have updated all relevant tables in the supplementary materials to follow a standardized format.
Point 5: Page 6. "The execution time of the whole pipeline, including the optimization, took around 6 hours per dataset on a machine with 16 GB RAM and 10-core Intel Xeon 4310 processors". The four datasets contained fundamentally different numbers of cases. It would be helpful to include specific running times for the four datasets.
Response: The execution time of the whole pipeline, including the optimization, took around 6 hours per for the LLN and SLN datasets and around 3 hours for the MBC and CHE dataset
We appreciate your suggestion to provide dataset-specific execution times given the different sample sizes across our validation datasets. We have updated the manuscript to include the specific running times for each dataset.
The following updates were made:
- Manuscript line 202 (Materials and Methods -> Statistical analysis) - Updated the total amount of combinations
Point 6: It is not clear how (in which process) the discretization step was included in the radiomics calculation. The discretization could be an absolute or relative method (fixed bin size or fixed bin number), as defined in https://doi.org/10.48550/arXiv.1612.07003. It has been demonstrated that various quantization methods can significantly influence the calculated radiomic/textural features and, more importantly, the correlational relationships between other variables.
Response: Thank you for raising this important methodological concern regarding discretization methods and their potential impact on radiomic feature calculations. We appreciate your reference to the relevant literature highlighting how different quantization approaches can significantly influence feature values and correlational relationships.
In our study, as feature extraction was out of the scope of our work, we started with CSV files containing pre-extracted and preprocessed radiomic features, to make RadiomiX as extraction-agnostic as possible. The datasets we utilized employed various extraction methodologies: some features were extracted using a stated 25 HU binning approach (LLN,SLN), while others did not specify the exact discretization methods used (MBC, CHE). Additionally, the features were extracted using different software platforms including TexLab 2.0 (developed in MATLAB 2015b, LLN, SLN), LifeX (MBC), and pyradiomics (CHE), and encompassed different feature types and orders. This approach was intentionally chosen to demonstrate RadiomiX's capability to work with heterogeneous radiomic data extracted using various methodologies, reflecting the real-world scenario where researchers may need to work with datasets that have been processed using different extraction pipelines and discretization strategies. We acknowledge that this introduces some methodological variability, but it also demonstrates the practical utility of our framework in diverse research contexts.
Point 7: The specific function form of the radiomics and texture parameters must correspond to the types defined/recommended in the IBSI standard. Did the implementation follow the function definitions in the recommendation (https://doi.org/10.48550/arXiv.1612.07003)? Were all the suggested functions of the IBSI implemented in RadiomiX?
Response: We appreciate this important question regarding adherence to IBSI standards for radiomic feature definitions. As mentioned in our response to Point 6, RadiomiX operates on pre-extracted radiomic features provided in CSV format rather than performing feature extraction itself. Therefore, RadiomiX does not directly implement the IBSI-recommended function definitions, as it is designed to be extraction-agnostic and work with features that have already been computed using various extraction software platforms.
The compliance with IBSI standards varies among the datasets included in our paper. Specifically, the MBC and CHE datasets utilized radiomic features that were IBSI-compliant, as they were extracted using PyRadiomics and LifeX respectively, with the CHE study explicitly stating their adherence to IBSI standards. In contrast, the LLN and SLN datasets used features extracted with TexLab 2.0, which can potentially be IBSI-compliant but does not necessarily guarantee adherence to these standards. This variation in our validation datasets actually demonstrates RadiomiX's practical utility in real-world scenarios where researchers may need to work with both IBSI-compliant and legacy radiomic datasets.
Point 8: Page 5. "For each dataset, a RadiomiX pipeline was built for testing the predefined 532 model combinations and evaluating them over 10 train-test splits and 10-fold CV." Please briefly describe why there are a total of 532 different model combinations.
Response: We appreciate your request for clarification regarding the calculation of model combinations. Upon reviewing this section, we identified an inconsistency in our reported numbers that requires correction. The figure of 532 model combinations referenced on page 5 was based on an earlier version of our algorithm configuration with a different set of predefined feature selection thresholds, which also corresponds to the feature range (1-100) mentioned in that analysis.
In our current and final implementation, we refined our approach to focus on lower feature counts more appropriate for the typically small sample sizes in radiomic studies. The updated and correct number of model combinations is 4,704, which is systematically calculated as: 4 scaling methods × 12 oversampling methods × 14 feature selection methods × 7 classifiers = 4,704 total combinations. We have corrected this discrepancy throughout the manuscript to ensure consistency and accuracy in reporting our comprehensive evaluation strategy.
The following updates were made:
- Manuscript line 114 (Materials and Methods -> Algorithm Description) - Changed the number of predefined features to 1,2,5,10,15,25.
- Manuscript line 171-172 (Materials and Methods -> Statistical analysis) - Updated the total amount of combinations and added the calculation explanation
- Manuscript line 179 (Materials and Methods -> Statistical analysis) - Updated the total amount of combinations
Point 9: Table 2 shows the results of the comparative analysis between the best models selected by RadiomiX and the originally published data from the four distinct radiomics datasets. For basic comparison purposes, it would be helpful to see the AUC and accuracy results calculated by RadiomiX, which were determined using exactly the same steps/methods (scalers, feature selection, classifier, and related hyperparameters) as in the original publications
Response: Thank you for your suggestion. To address this, we have already implemented the approach you suggest in our analysis. Specifically, we used RadiomiX to replicate the original published pipelines for each dataset according to their reported descriptions, including their exact number of data splits, feature selection methods, classifiers, and performance metrics. This allowed us to obtain comprehensive performance distributions for the original methodological approaches, providing a more robust basis for comparison than relying solely on single-point estimates from the original publications.
Following your suggestion, we have now clarified this approach in our statistical analysis section.
The following updates were made:
- Manuscript line 185-186 (Materials and Methods -> Statistical analysis) - added “using RadiomiX”.
Point 10: The last table in the Supplemental material S2 presents the "optimal pipelines' settings and hyperparameters" for the four different datasets. In the case of the MBC and the CHE, the classifier and feature selection methods were the same; however, the hyperparameters differed. Additionally, the authors emphasized that optimizing hyperparameters was outside the scope of this study. In this regard, why did the hyperparameters differ between the MBC and CHE datasets?
Response: You have made an excellent observation regarding the apparent contradiction between our statement about hyperparameter optimization being outside the scope and the differing hyperparameters between the MBC and CHE datasets.
When we stated that optimizing hyperparameters was outside the scope of this study, we meant that we did not conduct an in-depth, comprehensive exploration of hyperparameter optimization as extensively as we did for other aspects of our work. However, since our implementation utilized grid search functionality, we incorporated basic hyperparameter tuning within our framework. This was particularly important because many default settings from standard algorithms (such as Random Forest and XGBoost with n_estimators=100) are often too large and inappropriate for small radiomic datasets like MBC (with only 124 samples).
Rather than using default parameters that could lead to overfitting, we provided reasonable hyperparameter ranges without focusing on their true optimization or analyzing trends to find the optimal settings for each specific case. We employed a quick and iterative approach with a limited set of options that would be more relevant and appropriate for radiomic datasets. This explains why the hyperparameters differed between MBC and CHE datasets - they were selected from our predefined ranges based on the grid search results for each specific dataset. Future work should indeed include much deeper analysis of optimal hyperparameter ranges specifically tailored for small sample size, high-dimensional radiomic datasets.
Round 2
Reviewer 1 Report
Comments and Suggestions for Authors
I recommend to add some description as that attached in the word file realted to feature selection and classifiers with their equations to enrich the paper with full understanding information
Author Response
We thank the reviewer for the suggestion to include detailed descriptions and equations related to feature selection methods and classifiers, as outlined in the attached file. While we recognize the value of such information for readers seeking deeper methodological insight, we believe that incorporating it into the main manuscript would detract from the central focus of the paper, which is the development and validation of the RadiomiX pipeline.
To maintain clarity and emphasize our core message, we have chosen to provide these technical details in the supplementary material, where they remain accessible to interested readers. Furthermore, full documentation of the feature selection techniques, classifiers, and their implementation, including equations and code, is available in the RadiomiX GitHub repository [https://github.com/HarelKotler/RadiomiX].
We hope this approach strikes an appropriate balance between completeness and readability, and we trust the reviewer will understand our rationale.
Reviewer 2 Report
Comments and Suggestions for Authors
I thank the authors for incorporating my suggested changes; in addition, the manuscript's clarity has improved significantly. However, I would like to make one minor comment.
Based on the author's answers in points 6, 7, and 10, it would be worthwhile to supplement the discussion, as their content may also provide helpful information for later readers.
Author Response
I thank the authors for incorporating my suggested changes; in addition, the manuscript's clarity has improved significantly. However, I would like to make one minor comment.
Based on the author's answers in points 6, 7, and 10, it would be worthwhile to supplement the discussion, as their content may also provide helpful information for later readers.
Response: We sincerely thank the reviewer for the positive feedback and for acknowledging the improved clarity of the manuscript. Regarding the suggestion to integrate elements from our responses to points 6, 7, and 10 into the Discussion section, we agree that these clarifications may benefit future readers. Accordingly, we have expanded the Discussion to include the relevant points.